# Newton-like Normal S-iteration under Weak Conditions

**Manoj K. Singh** [1] ◉**, Ioannis K. Argyros** [2,*] **and Arvind K. Singh** [3,*]

1 College of Technology, Sardar Vallabhbhai Patel University of Agriculture and Technology, Meerut 250110, India
2 Department of Mathematical Sciences, Cameron University, Lawton, OK 73505, USA
3 Department of Mathematics, Institute of Science, Banaras Hindu University, Varanasi 221005, India
* Correspondence: iargyros@cameron.edu (I.K.A.); aksingh9@gmail.com (A.K.S.)

**Abstract:** In the present paper, we introduced a quadratically convergent Newton-like normal S-iteration method free from the second derivative for the solution of nonlinear equations permitting $f'(x) = 0$ at some points in the neighborhood of the root. Our proposed method works well when the Newton method fails and performs even better than some higher-order converging methods. Numerical results verified that the Newton-like normal S-iteration method converges faster than Fang et al.'s method. We studied different aspects of the normal S-iteration method regarding the faster convergence to the root. Lastly, the dynamic results support the numerical results and explain the convergence, divergence, and stability of the proposed method.

**Keywords:** Newton's method; normal S-iteration; weak condition; simple root; order of convergence

**MSC:** 65H04; 65H05; 30C15; 37N30





## 1. Introduction

In this work, we propose a Newton-like normal S-iteration method for solving nonlinear algebraic and transcendental equations of the following form [1–3]:

$$f(x) = 0. \tag{1}$$

Newton's method [4,5] is a basic method for solving (1), which converges to the root quadratically under some conditions. Newton's method is defined as follows:

$$x_{n+1} = x_n - \frac{f(x_n)}{f'(x_n)}, \quad n = 0, 1, 2, \ldots. \tag{2}$$

Although Newton's method is the most important known and the most basic used method to solve Equation (1), some weaknesses of Newton's method [6–12] are as follows:

(i) It is only a second-order method;
(ii) The initial approximation should be near the root;
(iii) The denominator term of Newton's method must not be zero at the root or near the root.

To overcome these weaknesses, Wu [12] developed a quadratic convergent method in 2000, which is expressed as follows:

$$x_{n+1} = x_n - \frac{f(x_n)}{\lambda_n f(x_n) + f'(x_n)}, \quad n = 0, 1, 2, \ldots, \tag{3}$$

where $|\lambda_n| \in (0, \infty)$. Fang et al. [11] studied a method in 2008, defined as follows:

$$\begin{cases} y_n = x_n + \frac{f(x_n)}{\lambda_n f(x_n) + f'(x_n)}, \\ x_{n+1} = y_n - \frac{f(y_n)}{\lambda_n f(x_n) + f'(x_n)}, \quad n = 0, 1, 2, \ldots, \end{cases} \tag{4}$$

where $|\lambda_n| \le 1$. They claimed that their method (4) is of cubic convergence. More precisely,

**Theorem 1** ([11]). *Let $f : I \subseteq \Re \to \Re$ be a function and assume that*
    *(L1) $x^* \in I$ is a simple zero of $f$;*
    *(L2) $f$ is three times differentiable on $I$;*
    *(L3) $\lambda_n f(x) + f'(x) \ne 0$, for all $x \in N(x^*)$ where $N(x^*)$ is a neighborhood of $x^*$. Then,*
*method (4) converges cubically to $x^*$.*

Recently, Wang and Liu [13] revealed that Fang et al.'s method given by (4) is only of second order. Wang and Liu [13] revised theorem 1 as follows:

**Theorem 2.** *Let $f : I \subseteq \Re \to \Re$ be a function and assume that*
    *(i) $x^* \in I$ is a simple zero of $f$;*
    *(ii) $f$ is three times differentiable on $I$;*
    *(iii) $\lambda_n f(x) + f'(x) \ne 0$, for all $x \in N(x^*)$, where $N(x^*)$ is a neighborhood of $x^*$. Then,*
*method (4) converges quadratically to $x^*$.*

More recently, Wang and Liu [13] modified method (4) for third-order convergence as follows:

$$\begin{cases} y_n = x_n + \dfrac{f(x_n)}{\lambda_n f(x_n) - f'(x_n)}, \\ x_{n+1} = y_n + \dfrac{f(y_n)}{\lambda_n f(x_n) - f'(x_n)}, & n = 0, 1, 2, \dots, \end{cases} \tag{5}$$

where $|\lambda_n| \le 1$ and it is equal to $-sign\big(f(x_n)f'(x_n)\big) min\{1, |f(x_n)|\}$. Under the above modification, Wang and Liu [13] settled the third-order convergence theorem as follows:

**Theorem 3.** *Let $f : I \subseteq \Re \to \Re$ be a function and assume that*
    *(W1) $x^* \in I$ is a simple zero of $f$;*
    *(W2) $f$ is three times differentiable on $I$;*
    *(W3) $\lambda_n f(x) - f'(x) \ne 0$ for all $x \in N(x^*)$, where $N(x^*)$ is a neighborhood of $x^*$.*
*Then, the iterative method (5) is cubically convergent.*

It is clear from condition *(W2)* of Theorem 3 that the sufficient condition for the convergence of method (5) to the zero of the function $f$ is that the third derivative of $f$ must exist. However, we often come across a situation in which the third-order derivative of the function does not exist, while $f$ has a zero in the interval $I$. Consider the function $f_1$ defined by

$$f_1(x) = x^{5/2} - \exp(x) + 1.$$

Here, $x^* = 0.0$. Note that $f_1(x^*) = 0$, and $f_1'''(x^*)$ does not exist. Hence, we observe that

(i)   Newton's method (2) can not be used;
(ii)  The method of Wang and Liu (5) does not satisfy the condition *(W2)* of Theorem 3. At this stage, the following question naturally arises: Is it possible to propose an iterative method for finding the solution of (1), when $f$ is not three times differentiable on $I$?

The objective of this work is to introduce a Newton-like normal S-iteration method for solving nonlinear Equation (1). Taking this into account, we describe a new method in which the second derivative of function $f$ is sufficient for convergence and is well comparable to third-order methods. For this purpose, we applied the normal S-iteration process to a second-order converging Newton-like method. The novelty of our proposed Newton-like normal S-iteration method is that when we compare it with other methods, the theoretical results remain the same, but the numerical results and dynamic results significantly improve. In the theoretical section, we show that due to having a quadratic converging method, it requires only second-order differentiability. In the numerical section, we verify the theoretical results with numerical examples and show that in spite of being a

second-order method, the present method is well comparable to the previously published second- and third-order methods. Furthermore, with sensitivity analysis, we determined the optimality conditions for the proposed method by finding the suitable values of $\lambda_n$ and $\beta_n$ in order to obtain the optimum results and also obtained the average number of iterations by performing several operations of the proposed method considering the 50 grid points. Lastly, We confirmed the theoretical and numerical results using the dynamic analysis of the proposed method. Thus, we plotted the fractal pattern graphs of the proposed method alongside those of the previously published methods to confirm the applicability of the proposed method. The results show that our method not only answers the research question affirmatively but also behaves very well in comparison to the third-order method developed by Wang and Liu [13].

The rest of this paper is arranged as follows: Section 2 presents preliminary results. In Section 3, we propose the new Newton-like normal S-iteration method and explain its convergence analysis. In Section 4, numerical examples are given to verify the theoretical results. Section 5 is related to the sensitivity analysis. Lastly, dynamic analysis supports the numerical and theoretical results in Section 6.

## 2. Preliminary

Let $x^*$ be a root of nonlinear Equation (1) and $f$ be a sufficiently differentiable function and $x_n \in N(x^*)$, where $N(x^*)$ is a neighborhood of $x^*$. Then, the numerical solution of (1) can be written as

$$f(x) = f(x_n) + \int_{x_n}^{x} f'(t)dt. \tag{6}$$

Approximating the integral by $(x - x_n)f'(x_n)$ with $x = x^*$ in (6), we obtain

$$0 \approx f(x_n) + (x^* - x_n)f'(x_n).$$

Therefore, a new approximation $x_{n+1}$ to $x^*$ can be written as (2). The Newton method (2) fails when the derivative of the $f$ becomes zero in the neighborhood of the root. On replacing $f'(x_n)$ in (2) by $f'(x_n) + \lambda_n f(x_n)$, we obtain an approximation $x_{n+1}$, as given in (3), which is the quadratically convergent method given by Wu [12].

## 3. New Newton-like Method and Its Theoretical Convergence Analysis

In this section, we introduce the new Newton-like normal S-iteration method and study its convergence analysis.

In [14], Sahu introduced a normal S-iteration process as follows:

**Definition 1.** *Let D be a nonempty convex subset of a normed space X and $T : D \to D$ be an operator. Then, for arbitrary $x_0 \in D$, the normal S-iteration process is defined by*

$$x_{n+1} = T\big((1 - \beta_n)x_n + \beta_n T(x_n)\big), \quad n = 0, 1, 2, \ldots,$$

*where the sequence $\beta_n \in (0, 1)$.*

There are many papers dealing with the S-iteration process and the normal S-iteration process in the literature. In [15], Sahu introduced a Newton-like method based on normal the S-iteration process as follows:

$$\begin{cases} x_{n+1} = y_n - \frac{f(y_n)}{f'(y_n)}, \\ y_n = (1 - \beta_n)x_n + \beta_n u_n, \\ u_n = x_n - \frac{f(x_n)}{f'(x_n)}, \quad n = 0, 1, 2, \ldots, \end{cases}$$

where the sequence $\beta_n \in (0, 1)$ and $f'(x)$ is the derivative of $f$ at point $x$.

We now introduce our new Newton-like normal S-iteration method for solving non-linear Equation (1), when $f'$ may be zero in the neighborhood of the root, as

$$\begin{cases} y_n = (1 - \beta_n)x_n + \beta_n G(x_n), \\ x_{n+1} = G(y_n), \quad n = 0, 1, 2, \ldots, \end{cases} \tag{7}$$

where

$$G(x_n) = x_n + \frac{f(x_n)}{\lambda_n f(x_n) - f'(x_n)}, \tag{8}$$

$\beta_n \in (0, 1)$ and $\lambda_n$ is a sequence in $\Re$, such that $|\lambda_n| \leq 1$. The parameter $\lambda_n$ is chosen in such a manner that both $\lambda_n f(x_n)$ and $-f'(x_n)$ have the same sign, and hence denominator is nonzero in Equation (8). For this purpose, we use the signum function as follows:

$$sign(x) = \begin{cases} 1, & if\ x \geq 0, \\ -1, & if\ x < 0. \end{cases}$$

The main result of this paper can now be established as follows:

**Theorem 4.** *Let $f : I \subseteq \Re \to \Re$ be a function and assume that*
*(i) $x^* \in I$ is a simple zero of $f$;*
*(ii) $f$ is two times differentiable on $I$;*
*(iii) $\lambda_n f(x) - f'(x) \neq 0$, for all $x \in N(x^*)$, where $N(x^*)$ is neighborhood of $x^*$ and $| \lambda_n |\leq 1$.*
*Then, the Newton-like normal S-iteration method defined by (7) is quadratically convergent locally to the zero of $f$.*

**Proof.** Let $x^* \in I$ be a simple zero of a function $f$, $e_n = x_n - x^*$ and $A_k = \left(\frac{1}{k!}\right) f^{(k)}(x^*) / f'(x^*)$. Using Taylor expansion about $x^*$ and using $f(x^*) = 0$, we obtain

$$f(x_n) = f'(x^*)\left[e_n + A_2 e_n^2 + A_3 e_n^3 + O(e_n^4)\right], \tag{9}$$

$$f'(x_n) = f'(x^*)\left[1 + 2A_2 e_n + 3A_3 e_n^2 + 4A_4 e_n^3 + O(e_n^4)\right]. \tag{10}$$

Now, from the above two equations, we obtain

$$\begin{aligned} f'(x_n) - \lambda_n f(x_n) &= f'(x^*)[1 + (2A_2 - \lambda_n)e_n + (3A_3 - \lambda_n A_2)e_n^2 + (4A_4 - \lambda_n A_3)e_n^3 \\ &+ O(e_n^4)] \end{aligned} \tag{11}$$

and from (9) and (11), we have

$$\frac{f(x_n)}{\lambda_n f(x_n) - f'(x_n)} = -e_n + (A_2 - \lambda_n)e_n^2 + \left(2A_2\lambda_n - \lambda_n^2 - 2A_2^2 + 2A_3\right)e_n^3 + O(e_n^4).$$

Using the above in (8), we obtain

$$G(x_n) = x^* + (A_2 - \lambda_n)e_n^2 + \left(2A_2\lambda_n - \lambda_n^2 - 2A_2^2 + 2A_3\right)e_n^3 + O(e_n^4). \tag{12}$$

Now, on using (12) in the first substep of (7), we get

$$y_n = x^* + (1 - \beta_n)e_n + \beta_n(A_2 - \lambda_n)e_n^2 + \beta_n\left(2A_2\lambda_n - \lambda_n^2 - 2A_2^2 + 2A_3\right)e_n^3 + O(e_n^4). \tag{13}$$

On expanding $f(y_n)$ and $f'(y_n)$ about $x_n$, we obtain

$$
\begin{aligned}
f(y_n) = \; & f'(x^*)\Big[(1-\beta_n)e_n + \big\{A_2(1-\beta_n)^2 + \beta_n(A_2 - \lambda_n)\big\}e_n^2 \\
& + \beta_n\big\{2A_2\lambda_n - \lambda_n^2 - 2A_2^2 + 2A_3 + 2A_2(1-\beta_n)(A_2 - \lambda_n)\big\}e_n^3 \\
& + O(e_n^4)\Big],
\end{aligned}
\tag{14}
$$

$$
\begin{aligned}
f'(y_n) = \; & f'(x^*)\Big[1 + 2A_2(1-\beta_n)e_n + \big\{3A_3(1-\beta_n)^2 + 2A_2\beta_n(A_2 - \lambda_n)\big\}e_n^2 \\
& + \beta_n\big\{2A_2\big(2A_2\lambda_n - \lambda_n^2 - 2A_2^2 + 2A_3\big) + 6A_3(1-\beta_n)(A_2 - \lambda_n)\big\}e_n^3 \\
& + O(e_n^4)\Big].
\end{aligned}
\tag{15}
$$

Now, from (14) and (15), we have

$$
\begin{aligned}
\lambda_n f(y_n) - f'(y_n) = \; & f'(x^*)\big[-1 + (1-\beta_n)(\lambda_n - 2A_2)e_n \\
& + \lambda_n\big\{(1-\beta_n)^2(A_2 - 3A_3) + \beta_n(A_2 - \lambda_n)(1 + 2A_2)\big\}e_n^2 \\
& + O(e_n^3)\big].
\end{aligned}
\tag{16}
$$

Furthermore, from (14) and (16), we have

$$
\begin{aligned}
\frac{f(y_n)}{\lambda_n f(y_n) - f'(y_n)} = \; & -(1-\beta_n)e_n + \{\lambda_n - 3A_2 - \beta_n(\lambda_n - 5A_2) \\
& + \beta_n^2(\lambda_n - 3A_2)\big\}e_n^2 + O(e_n^3).
\end{aligned}
\tag{17}
$$

With the help of (17), the second equation of (7) becomes

$$
x_{n+1} = x^* + \Big\{\lambda_n - 3A_2 - \beta_n(2\lambda_n - 6A_2) + \beta_n^2(\lambda_n - 3A_2)\Big\}e_n^2 + O(e_n^3)
$$

$$
\Rightarrow e_{n+1} = Ce_n^2 + O(e_n^3)
\tag{18}
$$

where $C = \lambda_n - 3A_2 - \beta_n(2\lambda_n - 6A_2) + \beta_n^2(\lambda_n - 3A_2)$. $\quad\square$

Hence, the Newton-like normal S-iteration method proposed in (7) has second-order convergence.

## 4. Numerical Analysis

In this section, we present some numerical tests to show the applicability of the proposed method by considering two categories of functions, namely (i) functions that are differentiable three times, and (ii) functions that are differentiable only two times. Numerical computations were carried out in MATLAB 2007 and the stopping criteria were taken as (i) $|f'(x_k)| \leq \varepsilon$, (ii) $|x_k - x_{k-1}| \leq \varepsilon$, where $\varepsilon = 10^{-15}$. We applied the Newton-like normal S-iteration method for the following three values of $\lambda_n$:

(i) $|\lambda_n| = 0.5$;
(ii) $|\lambda_n| = 1.0$;
(iii) $\lambda_n = -sign(f(x_n)f'(x_n))min\{1, |f(x_n)|\}$ ($\lambda_n$ is taken as in Wang and Liu [13]).

(i) *Functions with third-order differentials:*

Here, we consider the examples taken by Wang and Liu [13] as follows:

$$
F_1(x) = x\sin x + \cos x - 0.6, \quad x^* = -2.54623173142842,
$$

$$
F_2(x) = x^3 - 2x^2 + x - 1, \quad x^* = 1.75487766624669,
$$

$$
F_3(x) = \ln x, \quad x^* = 1.0000,
$$

$$F_4(x) = \arctan x, \quad x^* = 0.0000,$$

$$F_5(x) = x + 1 - \exp(\sin x), \quad x^* = 1.69681238680975,$$

$$F_6(x) = x \exp(-x^2) - (\sin x)^2 + 3\cos x + 5, \quad x^* = -1.20764782713092,$$

$$F_7(x) = 10x \exp(-x^2) - 1. \quad x^* = 1.67963061042845.$$

For the two values of $\lambda_n = 0.5$ and $\lambda_n$ as indicated in Wang's method, we considered $\beta_n = 0.5$ and 0.9, as shown in Table 1. When starting with the same initial points as in Wang and Liu [13] in all test problems, we observe that for both values of $\lambda_n$, our normal S-iteration method takes less number of iterations than the method of Wang and Liu method [13] for the value of $\beta_n = 0.9$. Thus, in spite of being a second-order convergence method, it performs better than the third-order method of Wang and Liu [13]. Furthermore, It may be noted that in all test problems, the classical Newton's method either fails or diverges in most cases. In Table 1, $F$, $D$, and $NC$ denote the failure of the method, the divergence of the method, and not converging to the desired root, respectively.

**Table 1.** Functions for which third-order differentials exist.

| $f(x)$ | $x_0$ | Newton's Method | Wang and Liu's Method | Normal S-Iteration Method | | | |
| --- | --- | --- | --- | --- | --- | --- | --- |
| | | | | $|\lambda_n| = 0.5$ | | $\lambda_n$ as Wang and Liu | |
| | | | | $\beta_n = 0.5$ | $\beta_n = 0.9$ | $\beta_n = 0.5$ | $\beta_n = 0.9$ |
| $F_1$ | 0 | F | 5 | 7 | 5 | 5 | 4 |
| | −4 | 6 | 5 | 5 | 4 | 6 | 5 |
| $F_2$ | 1 | F | 7 | 5 | 5 | 5 | 4 |
| | 3 | 7 | 6 | 6 | 6 | 6 | 5 |
| $F_3$ | 5 | D | 5 | 5 | 4 | 7 | 6 |
| | 2 | 6 | 4 | 3 | 3 | 5 | 4 |
| $F_4$ | 3 | D | 4 | 5 | 4 | 5 | 4 |
| | −1 | 5 | 3 | 4 | 3 | 4 | 3 |
| $F_5$ | 4 | NC | 6 | 6 | 5 | 7 | 6 |
| | 2 | 5 | 4 | 4 | 4 | 4 | 4 |
| $F_6$ | 0.73 | D | 8 | 6 | 4 | 8 | 4 |
| | −3 | 23 | 15 | 11 | 9 | 11 | 9 |
| $F_7$ | 0.7 | D | 5 | 4 | 4 | 4 | 4 |
| | 2 | 6 | 4 | 4 | 3 | 4 | 3 |

(ii) *Functions that are differentiable only two times*

We considered the following real functions from $I \subset \Re \to \Re$, and the results are shown in Table 2:

$$f_1(x) = x^{\frac{5}{2}} - \exp x + 1, \quad x^* = 0.0,$$

$$f_2(x) = x^4 \sin\frac{1}{x}, x \neq 0, \quad x^* = 0.31830988618379 (x_0 = 1),$$

$$x^* = 0.106103295394597 (x_0 = 0.1),$$

$$f_3(x) = x^{\frac{7}{3}} \sin x, \quad x^* = 0.0,$$

$$f_4(x) = (x-2)^{\frac{7}{3}} - x^3 + 3x^2 - 2, \quad x^* = 2.475200396019297,$$

$$f_5(x) = x^{\frac{7}{3}} \exp x, \quad x^* = 0.0,$$

$$f_6(x) = (x+2)^{\frac{5}{2}} + \exp x - 1, \quad x^* = -1.142466838767107.$$

**Table 2.** Functions for which third-order differential does not exist.

| $f(x)$ | $x_0$ | Newton Method | Fang et al. Method | Normal S-Iteration Method | | | | | |
|---|---|---|---|---|---|---|---|---|---|
| | | | | $\lambda_n$ as Wang and Liu | | $|\lambda_n| = 0.5$ | | $|\lambda_n| = 1$ | |
| | | | | $\beta_n = 0.9$ | $\beta_n = 0.5$ | $\beta_n = 0.9$ | $\beta_n = 0.5$ | $\beta_n = 0.9$ | $\beta_n = 0.5$ |
| $f_1$ | 0.5 | F | 7 | 3 | 4 | 3 | 4 | 4 | 5 |
| $f_2$ | 1.0 | 9 | 9 | 6 | 7 | 6 | 7 | 6 | 8 |
| | 0.1 | 5 | 5 | 3 | 4 | 3 | 4 | 3 | 4 |
| $f_3$ | 0.3 | 85 | 58 | 47 | 60 | 47 | 60 | 47 | 60 |
| | 1.0 | 88 | 61 | 49 | 62 | 49 | 62 | 49 | 62 |
| $f_4$ | 2.0 | F | 9 | 4 | 5 | 5 | 4 | 4 | 4 |
| $f_5$ | 1.0 | 89 | 43 | 33 | 42 | 33 | 42 | 33 | 43 |
| $f_6$ | −2.0 | 10 | F | 3 | 4 | 5 | 4 | 3 | 4 |

As we know from the condition *(W2)* of Theorem 3, the cubically convergent method of Wang and Liu will converge to the root only if the third-order differential of the function exists in the neighborhood of the root. Hence, Wang's method is no longer applicable in this case. Therefore, we compared the present method with quadratically convergent same-order Newton's method and Fang et al.'s method [11] for different values of $\lambda_n$ and $\beta_n$ ($\lambda_n = 0.5, \lambda_n = 1, \lambda$ as in Wang and Liu [13] and $\beta_n = 0.5, \beta_n = 0.9$) in Table 2. In all the test problems, for all values of $\lambda_n$ and $\beta_n$, we can see that the present new Newton-like normal S-iteration method is always taking less number of iterations, except for example 3 (case $\beta_n = 0.5$), in comparison to other quadratically convergent methods. Hence, we conclude that the present method is more effective, robust, and stable.

## 5. Sensitivity Analysis

### 5.1. The Behavior of Normal S-Iteration Method for Different Values of $\lambda_n$ and $\beta_n$

We took the function $F_6(x) = x \exp(-x^2) - (\sin x)^2 + 3\cos x + 5$, ($x^* = -1.20764782713092$) to investigate the empirical behavior of the proposed normal S-iteration method for different values of $\lambda_n$ and $\beta_n$. The numerical results in Table 3 show that when starting with the initial approximations 0.73 and −3.0, the proposed method is not significantly affected due to the variation in the value of $\lambda_n$, but the value of $\beta_n$ plays a crucial role as its different values are considered in the interval (0, 1). Thus, we can see that with the values of $\beta_n$ ranging from 0.1 to 0.9, the optimum value of $\beta_n$ is found to be 0.9 for which the proposed method is taking the least number of iterations.

**Table 3.** The proposed method for different values of $\lambda_n$ and $\beta_n$.

| $f(x)$ | $x_0$ | $\beta_n$ | Normal S-Iteration Method | | |
|---|---|---|---|---|---|
| | | | $|\lambda_n| = 0.5$ | $|\lambda_n| = 1$ | $\lambda_n$ as Wang and Liu |
| | | 0.1 | 13 | 9 | 9 |
| | | 0.3 | 7 | 8 | 8 |
| | 0.73 | 0.5 | 6 | 8 | 8 |
| | | 0.7 | 5 | 5 | 5 |
| | | 0.9 | 4 | 4 | 4 |
| $F_6$ | | | | | |
| | | 0.1 | 14 | 15 | 14 |
| | | 0.3 | 12 | 13 | 13 |
| | −3.0 | 0.5 | 11 | 11 | 11 |
| | | 0.7 | 10 | 10 | 10 |
| | | 0.9 | 8 | 9 | 9 |

### 5.2. Normal-S Iteration Method with Variable Value of $\beta$

We considered the two sequence of $\beta_n$ as $\beta_n^1 = 0.1 + 1/2(n + 2)$ and $\beta_n^2 = 1 - 1/2(n + 2)$ to solve the following two test functions using the proposed method:

$$(a) \quad F_1(x) = x \sin x + \cos x - 0.6, \quad x^* = -2.54623173142842$$

$$(b) \quad f_2(x) = x^4 \sin(1/x), \quad x^* = 0.31830988618379.$$

We observe from Table 4 that the second sequence $\beta_n^2 = 1 - 1/2(n+2)$ is taking fewer iterations in comparison to the first sequence $\beta_n^1 = 0.1 + 1/2(n+2)$ in converging to the root for both test functions. Hence, we conclude that the sequence that converges near 1, i.e., $\beta_n^2 = 1 - 1/2(n+2)$, gives the faster convergence to the root.

**Table 4.** Normal-S iteration method with a variable value of $\beta$.

| $f(x)$ | Normal S-Iteration for Sequence $\beta_n^1$ | | Normal S-Iteration for Sequence $\beta_n^2$ | |
| --- | --- | --- | --- | --- |
| | $\|\lambda_n\| = 0.5$ | $\lambda_n$ as Wang and Liu | $\|\lambda_n\| = 0.5$ | $\lambda_n$ as Wang and Liu |
| $F_1(x)$ | $-4.00000000000000$ | $-4.00000000000000$ | $-4.00000000000000$ | $-4.00000000000000$ |
| | $-3.019890471239318$ | $-3.269614812666443$ | $-2.787748595141695$ | $-3.031336002398129$ |
| | $-2.647689829523139$ | $-2.830596759888509$ | $-2.550732240466982$ | $-2.602227130430227$ |
| | $-2.552574309373607$ | $-2.5972691293100047$ | $-2.546231963106547$ | $-2.546267106449917$ |
| | $-2.546259317314531$ | $-2.547305870288047$ | $-2.546231731428419$ | $-2.546231731433164$ |
| | $-2.546231731968219$ | $-2.546232155885697$ | | $-2.546231731428418$ |
| | $-2.546231731428418$ | $-2.546231731428486$ | | |
| | | $-2.546231731428418$ | | |
| $f_2(x)$ | $1.000000000000000$ | $1.000000000000000$ | $1.000000000000000$ | $1.000000000000000$ |
| | $0.690862097114279$ | $0.713251419170333$ | $0.588489366623379$ | $0.613537499145787$ |
| | $0.500158984920628$ | $0.506202917231944$ | $0.388129276855868$ | $0.391429495638206$ |
| | $0.391718592801076$ | $0.390681052832476$ | $0.323527870651833$ | $0.323501217147855$ |
| | $0.338547374719877$ | $0.337061756299449$ | $0.318314259700137$ | $0.318313848040219$ |
| | $0.320626856711258$ | $0.320222652750527$ | $0.318309886184780$ | $0.318309886184561$ |
| | $0.318346374376978$ | $0.318333591884421$ | $0.318309886183791$ | $0.318309886183791$ |
| | $0.318309895590689$ | $0.318309889954683$ | | |
| | $0.318309886183791$ | $0.318309886183791$ | | |

### 5.3. Average Number of Iterations in Normal-S Iteration Method

Tables 5 and 6 show the average number of iterations denoted by ANI for 50 tests conducted with different values of $\beta_n$ [6]. For this purpose, we considered the following two test functions, which are three times differentiable:

**Example 1.** $F_2(x) = x^3 - 2x^2 + x - 1.0 = 0$.
It has root $x^* = 1.75487766624669$. We took the initial approximations in the grid as follows: $x_0 = 0.25 + ih, i = 1, \ldots, 50$ and $h = 0.03$ (see Table 5). The allowed error is $10^{-14}$.

**Example 2.** $F_6(x) = x \exp(-x^2) - (\sin x)^2 + 3 \cos x + 5$.
It has root $x^* = -1.20764782713092$. We took the initial approximations of $x_0$ in the grid as follows: $x_0 = -2.0 + ih, i = 1, \ldots, 50$, and $h = 0.03$ (see Table 6). The allowed error is $10^{-14}$.

**Table 5.** The average number of iterations in normal-S iteration method.

| $\beta$ | The average Number of Iterations (ANI) in Normal-S Iteration Method | | |
| --- | --- | --- | --- |
| | $\|\lambda_n\| = 0.5$ | $\|\lambda_n\| = 1$ | $\lambda_n$ as Wang and Liu |
| 0.1 | 5.340000 | 5.080000 | 5.100000 |
| 0.2 | 5.040000 | 4.920000 | 4.920000 |
| 0.3 | 4.800000 | 4.720000 | 4.600000 |
| 0.4 | 4.360000 | 4.480000 | 4.420000 |
| 0.5 | 4.240000 | 4.300000 | 4.280000 |
| 0.6 | 4.100000 | 4.080000 | 4.140000 |
| 0.7 | 3.800000 | 3.760000 | 3.640000 |
| 0.8 | 3.700000 | 3.600000 | 3.540000 |
| 0.9 | 3.620000 | 3.300000 | 3.340000 |

**Table 6.** The average number of iterations (ANI) in normal-S iteration method.

| $\beta$ | Average Number of Iterations in Normal-S Iteration Method | | |
|---|---|---|---|
| | $|\lambda_n| = 0.5$ | $|\lambda_n| = 1$ | $\lambda_n$ as Wang and Liu |
| 0.1 | 5.725490 | 5.411765 | 5.333333 |
| 0.2 | 5.411765 | 5.196078 | 5.137255 |
| 0.3 | 5.176471 | 4.941176 | 4.882353 |
| 0.4 | 4.980392 | 4.823529 | 4.764706 |
| 0.5 | 4.705882 | 4.666667 | 4.607843 |
| 0.6 | 4.431373 | 4.549020 | 4.450980 |
| 0.7 | 4.254902 | 4.352941 | 4.294118 |
| 0.8 | 3.764706 | 4.137255 | 4.058824 |
| 0.9 | 3.803922 | 3.764706 | 3.666667 |

*5.4. Convergence Behavior of the Methods of Newton and Fang et al. and the Present Method*

The convergence behavior of Newton's method, Fang et al.'s method [11], and the new Newton-like normal S-iteration method are shown in Figures 1–3. To study the convergence behavior, we took the test functions $f_2$, $f_3$, and $f_5$, and for each test function, we considered the three cases as follows:

**Case 1: The graph between function and root for $f_2$, $f_3$, and $f_5$**

Here, from Figure 1a, it is clear that for $x_0 = 1.0$, we have $f_2(x_0) = 0.841470984807896$. Starting with this initial approximation $x_0$, the value of $x_1$ for Newton's method, Fang et al.'s method [11], and the present method are 0.702195479022049, 0.677964714450141, and 0.576332178830878, respectively. Clearly, from Figure 1a, it can be inferred that the present method (red line) is better in its very first iteration among all three methods. After successive iterations, starting with $x_0 = 1.0$, the present method very rapidly converges to the root $x^* = 0.318309886183791$, as shown in the figure. Similarly, for the function $f_3$ in Figure 1b and the function $f_5$ in Figure 1c, we can see that the present method converges to the root $x^* = 0$ faster than others.

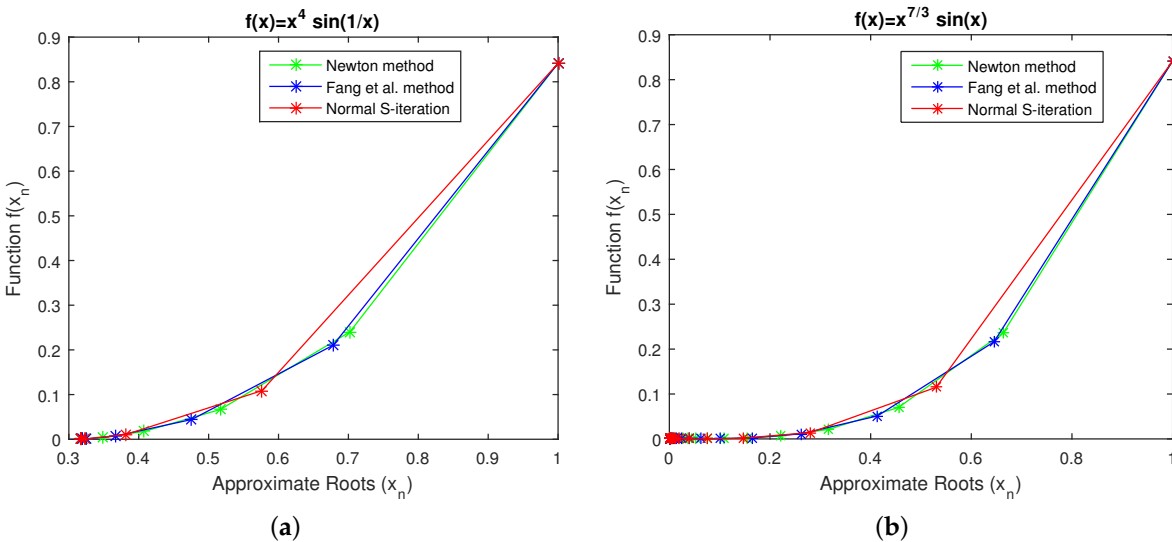

(**a**)　　　　　　　　　　　　　　　　(**b**)

**Figure 1.** *Cont.*

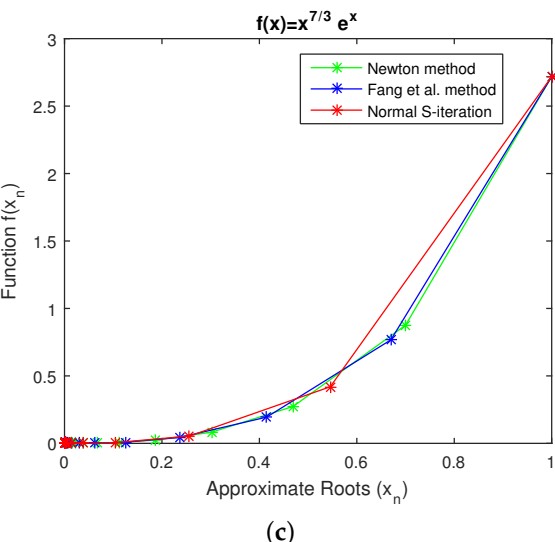

(**c**)

**Figure 1.** The graph using values of functions and roots: (**a**) Graph for $f_2$; (**b**) Graph for $f_3$; (**c**) Graph for $f_5$.

**Case 2: Graph of the number of iterations and roots for $f_2$, $f_3$, and $f_5$**

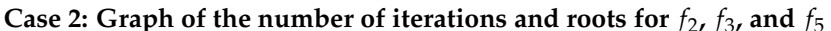

(**a**)

(**b**)

(**c**)

**Figure 2.** Graph using roots and number of iterations: (**a**) Graph for $f_2$; (**b**) Graph for $f_3$; (**c**) Graph for $f_5$.

For the function $f_3$, we have $f_3(x_0) = 0.841470984807896$ for $x_0 = 1.0$. It is clear from Figure 2b that when starting with the initial approximation $x_0$, Newton's method, Fang et al.'s method [11], and the present method converge to the root $x^* = 0.0$ in 88, 61, and 49 iterations, respectively. Hence, the new Newton-like normal S-iteration method takes fewer iterations among all the iterative methods. Similarly, we see the same pattern for $f_2$ and $f_5$ in Figure 2a,c.

**Case 3: Graph of number of iterations and functions for $f_2$, $f_3$, and $f_5$**

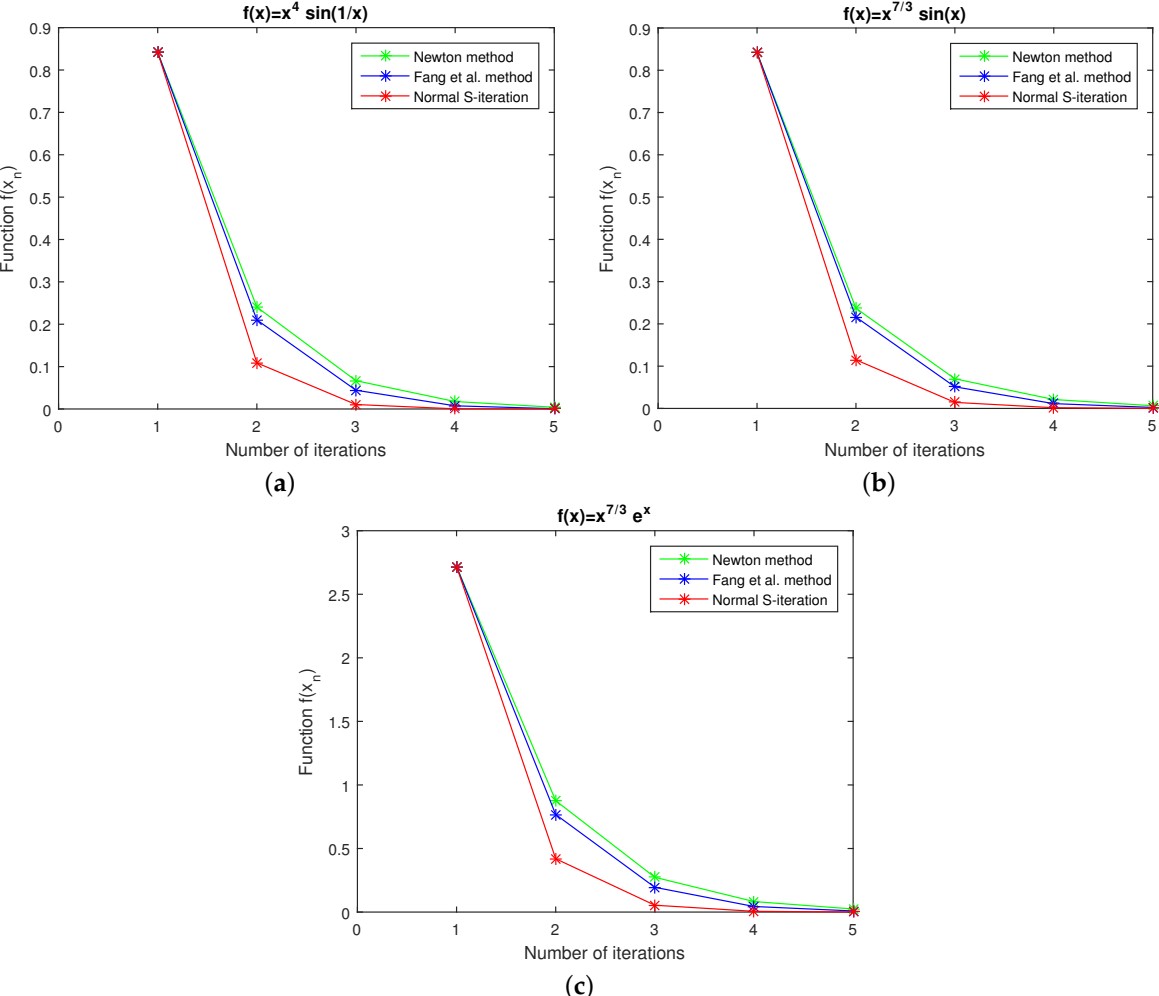

**Figure 3.** Graph for values of functions and numbers of iterations: (**a**) Graph for $f_2$; (**b**) Graph for $f_3$; (**c**) Graph for $f_5$.

In Figure 3c, we have $f_5(x_0) = 2.718281828459045$ for $x_0 = 1.0$. Starting with $x_0$, we can see from the graph that the value of the function $f_5$ in the present method becomes 0 in 33 iterations, while the Newton method and Fang et al.'s method [11] take 89 and 43 iterations, respectively, which shows that the present method converges to the root $x^* = 0.0$ faster than Newton's method and Fang et al.'s method. Figure 3a,b show the same results for functions $f_2$ and $f_3$, respectively.

## 6. Dynamic Analysis of Methods for Functions $f_1$, $f_2$, $F_1$, and $F_2$

Now, we define the following definitions but in the extended complex plane:

**Definition 2** ([10,16,17]). *Let us consider* $g : I \to \mathbb{C}$ *as a rational map on the Riemann sphere, where I is a subset of the complex numbers* $\mathbb{C}$. *Then, a point* $z_0$ *is said to be a fixed point of g if*

$$g(z_0) = z_0.$$

*Again, for any point* $z \in \mathbb{C}$, *the orbit of the point z can be defined as the set*

$$Orb(z) = \{z, g(z), g^2(z), \ldots, g^n(z), \ldots\}.$$

**Definition 3** ([10,16]). *A periodic point* $z_0$ *is said to be of period k if there exists a smallest positive integer k, i.e.,* $g^k(z_0) = z_0$.

**Remark 1.** *If* $z_0$ *is a periodic point of period k, then clearly, it is a fixed point for* $g^k$.

**Definition 4** ([10,16,17]). *Let* $z^*$ *be a zero of the function F. Then, the basin of attraction of the zero value* $z^*$ *is defined as the set of all initial approximations* $z_0$ *such that any numerical iterative method starting with* $z_0$ *converges to* $z^*$. *It can be written as*

$$B(z^*) = \{z_0 : z_{n+1} = g^n(z_0)\ converges\ \to\ z^*\}. \tag{19}$$

*Here,* $g^n$ *is any fixed point iterative method.*

**Remark 2.** *For example, in the case of Newton's method,*

$$z_{n+1} = g(z_n),$$

$$g(z_n) = z_n - \frac{F(z_n)}{F'(z_n)}, \quad n = 0, 1, 2, \ldots.$$

We can write the basin of attraction of the zero value $z^*$ for Newton's method as follows:

$$B(z^*) = \{z_0 : z_{n+1} = g^n(z_0)\ converges\ \to\ z^*\}.$$

**Definition 5** ([10,16,17]). *The Julia set of a nonlinear map* $g(z)$ *is denoted as* $J(g)$ *and is defined as a set consisting of the closure of its repelling periodic points [18]. The complement of Julia set* $J(g)$ *is called the Fatou set* $f(g)$.

**Remark 3.** *(i)    The Julia set of a nonlinear map may also be defined as the common boundary shared by the basins of roots, and the Fatou set may also be defined as the set that contains the basin of attraction.*

*(ii)    Sometimes, the Fatou set of a nonlinear map may also be defined as the solution space and the Julia set of a nonlinear map may also be defined as the error space;*

*(iii)    Fractals are very complicated phenomena that may be defined as self-similar unexpected geometric objects that are repeated at every small scale ([19]).*

We plotted the dynamics of the iterative methods for various functions. Then, we examined the theoretical and numerical results with the help of dynamic results. A dynamic study helps us to understand the convergence and stability of the methods [10]. We applied our method on a square $R \times R = [-5, 5] \times [-5, 5]$ of $700 \times 700$ points with a tolerance $|f(z_n)| < 5 \times 10^{-2}$ and a maximum of 30 iterations. For any function, if the sequence generated by the iterative methods with any initial point $z_0$ converges to a zero $z^*$ in the square, then point $z_0$ will lie in the basins of attraction of this zero, and we assign a fixed color to this point $z_0$ ([20]).

In the following, we describe the speed of convergence and dynamics of the considered methods under two cases for finding the complex roots of functions. In the first case, we plotted the speed of convergence and dynamics of Newton's method, Fang et al.'s method [11], and the proposed method for functions $f_1$, $f_2$ (for which the third-order derivative does not exist). In the second case, we studied the speed of convergence and dynamics of Newton's method, Wang and Liu's method [13], and the proposed method for functions $F_1$ and $F_2$ (for which the third-order derivative exists).

### 6.1. Functions for Which the Third-Order Derivative Does Not Exist

We took the following two functions, which are differentiable only two times.

$$f_1(x) = x^{\frac{5}{2}} - \exp x + 1, \quad x^* = 0.0,$$

$$f_2(x) = x^4 \sin(1/x), \quad x^* = 0.31830988618379.$$

For the function $f_1(x) = x^{\frac{5}{2}} - \exp x + 1$, $x^* = 0.0$, the dynamics and speed of convergence for various methods are shown in Figure 4a–c. It is clear from Figure 4 that the proposed method with $|\lambda_n| = 0.5$ and $\beta_n = 0.9$ generates a Fatou set with larger orbits and darker color and a Julia set with fewer fractal boundaries and less chaotic behavior. Newton's method shows some type of fractal boundaries and chaotic behavior in the middle and right side of Figure 4a. The dynamics of Fang et al.'s method [11] generates a Fatou set with smaller orbits but a larger Julia set and thus is considered the worst method.

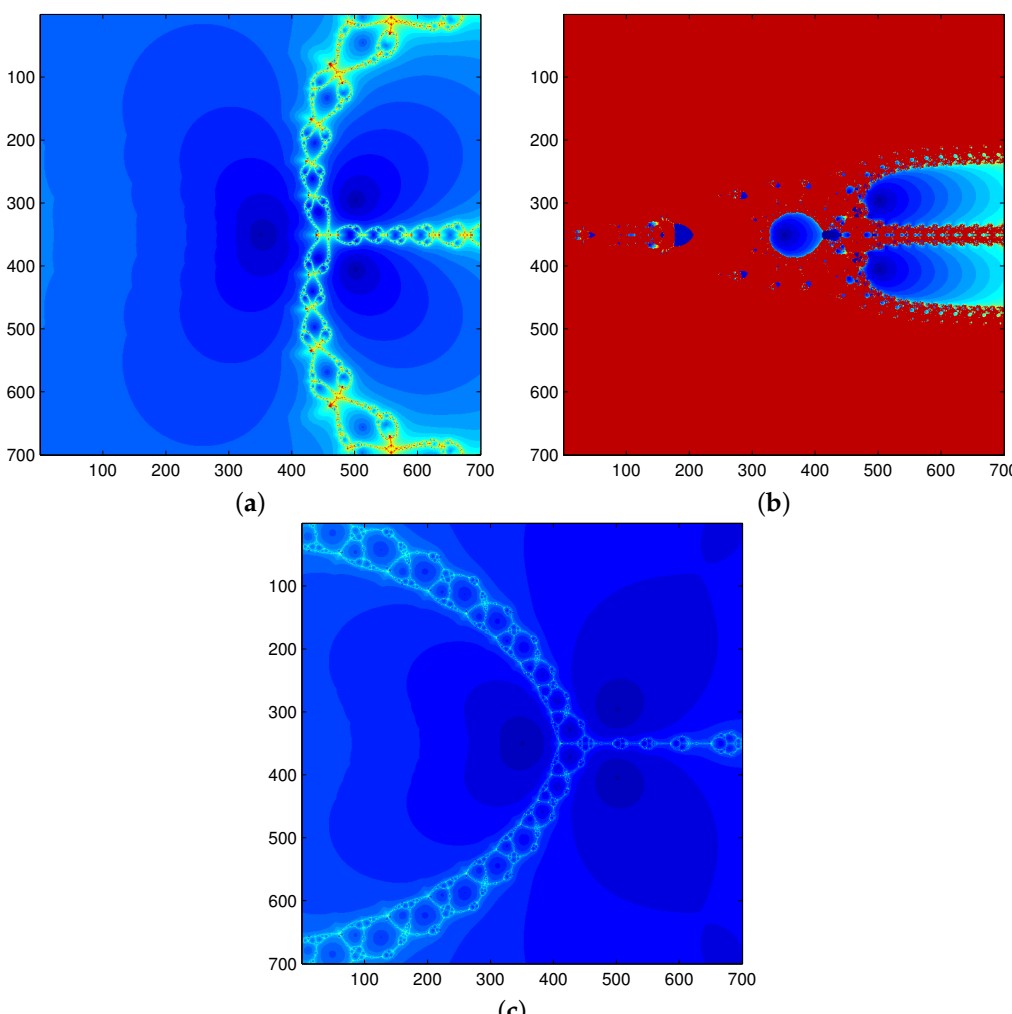

**Figure 4.** Dynamics of different methods for $f_1(x) = x^{\frac{5}{2}} - \exp x + 1$: (**a**) Newton's method; (**b**) Fang et al.'s method; (**c**) proposed method.

The dynamics and speed of convergence of Newton's method, Fang et al.'s method, and the proposed method for $f_2(x) = x^4 \sin(1/x)$, are plotted in Figure 5a, Figure 5b, and Figure 5c, respectively. Clearly, the fractal patterns graph of Newton's method has a large Julia set with fractal boundaries and chaotic behavior, whereas the proposed method and Fang et al.'s method [11] have a large Fatou set with basins, but both methods have some nonconverging regions, shown in the left side of the figures with red color.

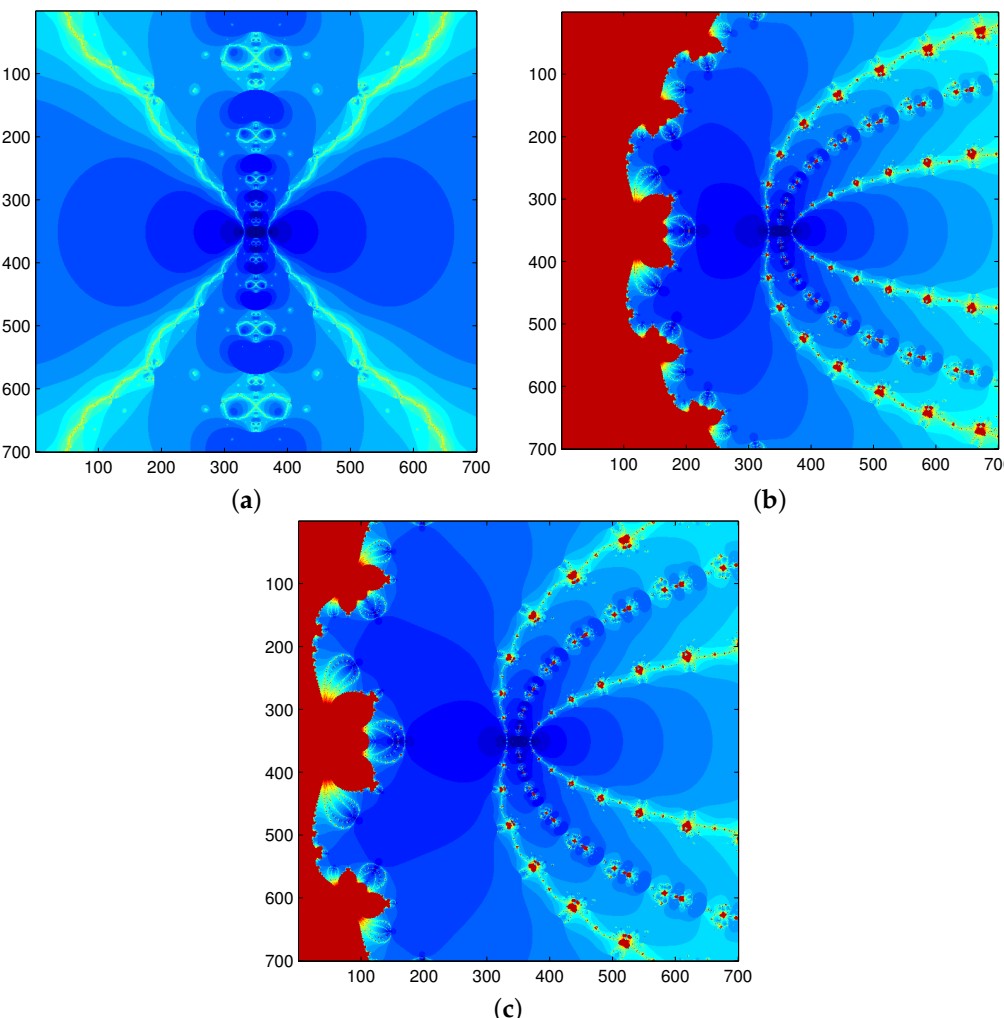

**Figure 5.** Dynamics of different methods for $f_2(x) = x^4 sin(1/x)$: (**a**) Newton's method; (**b**) Fang et al. method; (**c**) proposed method.

### 6.2. Functions for Which the Third-Order Derivative Exist

We took the following two functions, which are differentiable three times.

$$F_1(x) = x \sin x + \cos x - 0.6, \quad x^* = -2.54623173142842,$$

$$F_2(x) = x^3 - 2x^2 + x - 1, \quad x^* = 1.75487766624669.$$

For $F_1(x) = x \sin x + \cos x - 0.6$, the dynamics of Newton's method, Wang and Liu's method [13], and the proposed method can be seen in Figure 6a, Figure 6b, and Figure 6c, respectively. Here, Figure 6 shows that the proposed method with $|\lambda_n| = 0.5$ and $\beta_n = 0.9$ is the best method because of having a Fatou set with larger orbits and darker color and a Julia set with fewer fractal boundaries and less chaotic behavior. Wang and Liu's method [13] generates some type of chaotic behavior in the whole figure (Figure 6b). Newton's method generates a Fatou set with smaller orbits and a Julia set with less chaotic behavior with

reddish color in the middle of the figure (see Figure 6a). This is the reason why Newton's method takes several iterations and sometimes fails.

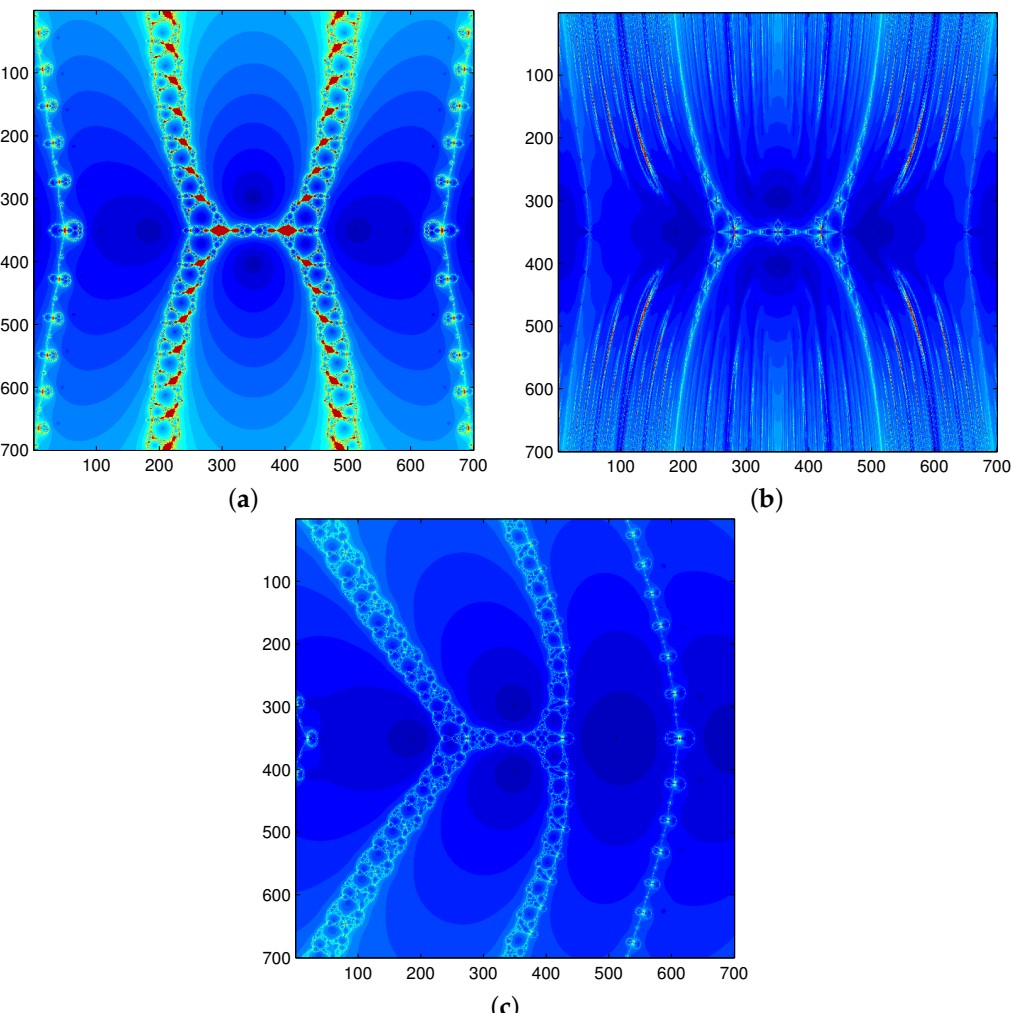

**Figure 6.** Dynamics of different methods for $F_1(x) = x \sin x + \cos x - 0.6$: (**a**) Newton's method; (**b**) Wang and Liu's method; (**c**) proposed method.

The dynamics of Newton's method, Wang and Liu's method, and the proposed method for function $F_2(x) = x^3 - 2x^2 + x - 1$ are shown in Figure 7a–c. The failure of Newton's method with the starting point $x_0 = 1.0$, as shown in Table 1, is proved in Figure 7a. The speed of the convergence of Newton's method and Wang and Liu's method [13] is slow with a fractal Julia set and chaotic behavior in comparison with the proposed method.

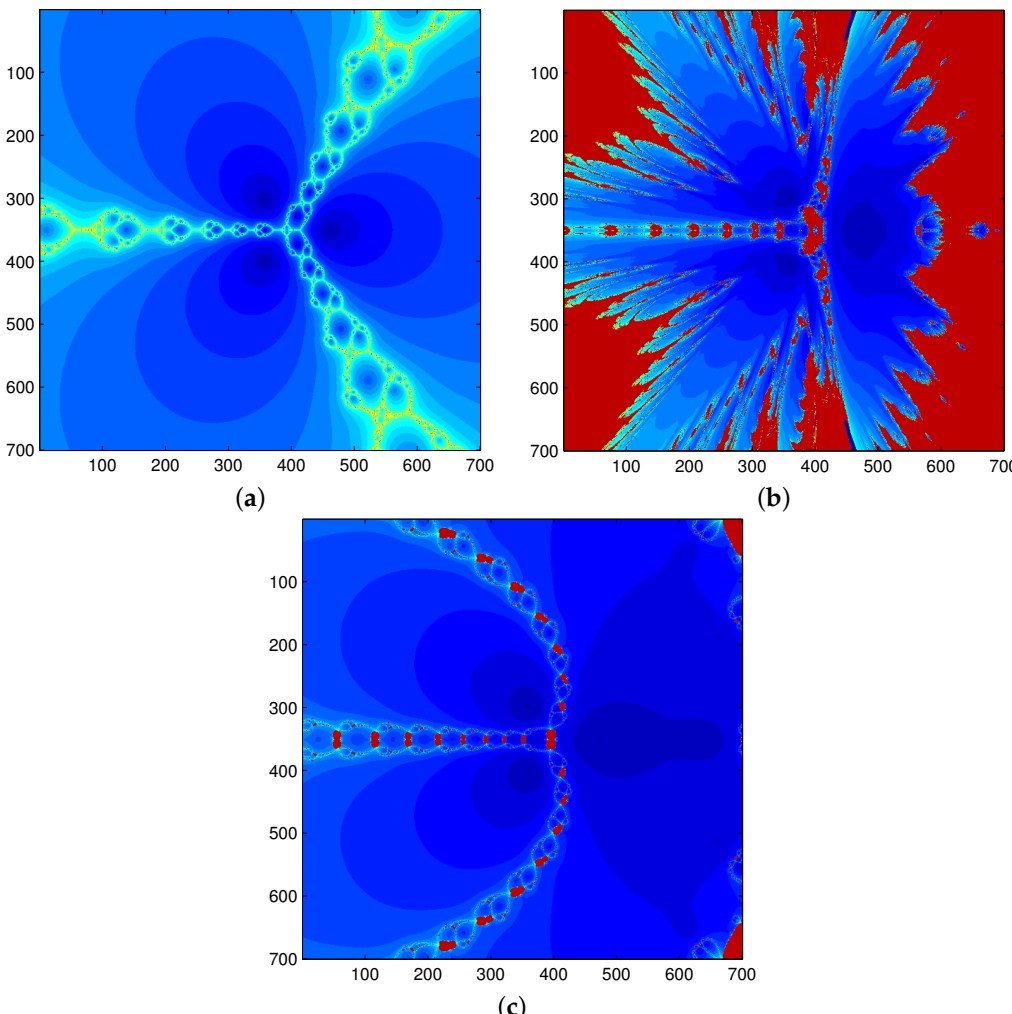

**Figure 7.** Dynamics of different methods for $F_2(x) = x^3 - 2x^2 + x - 1$: (**a**) Newton's method; (**b**) Wang and Liu's method; (**c**) proposed method.

*6.3. Dynamics of Proposed Method with Variable Value of $\beta$ for Example $F_2$*

We plotted the speed of convergence and dynamics of the proposed method with variable values of $\beta$ for $F_2(x) = x^3 - 2x^2 + x - 1$, $x^* = 1.75487766624669$. The results are shown in Figure 8. It is clear from the figure that the speed of the convergence of the proposed method increases with an increase in the value of $\beta$ as the Fatou set increases with larger orbits and a darker color. Moreover, for the value of $\beta = 0.9$, the speed of convergence is optimal with larger orbits and less chaotic behavior in comparison with $\beta = 0.1, 0.3, 0.5$, and $0.7$.

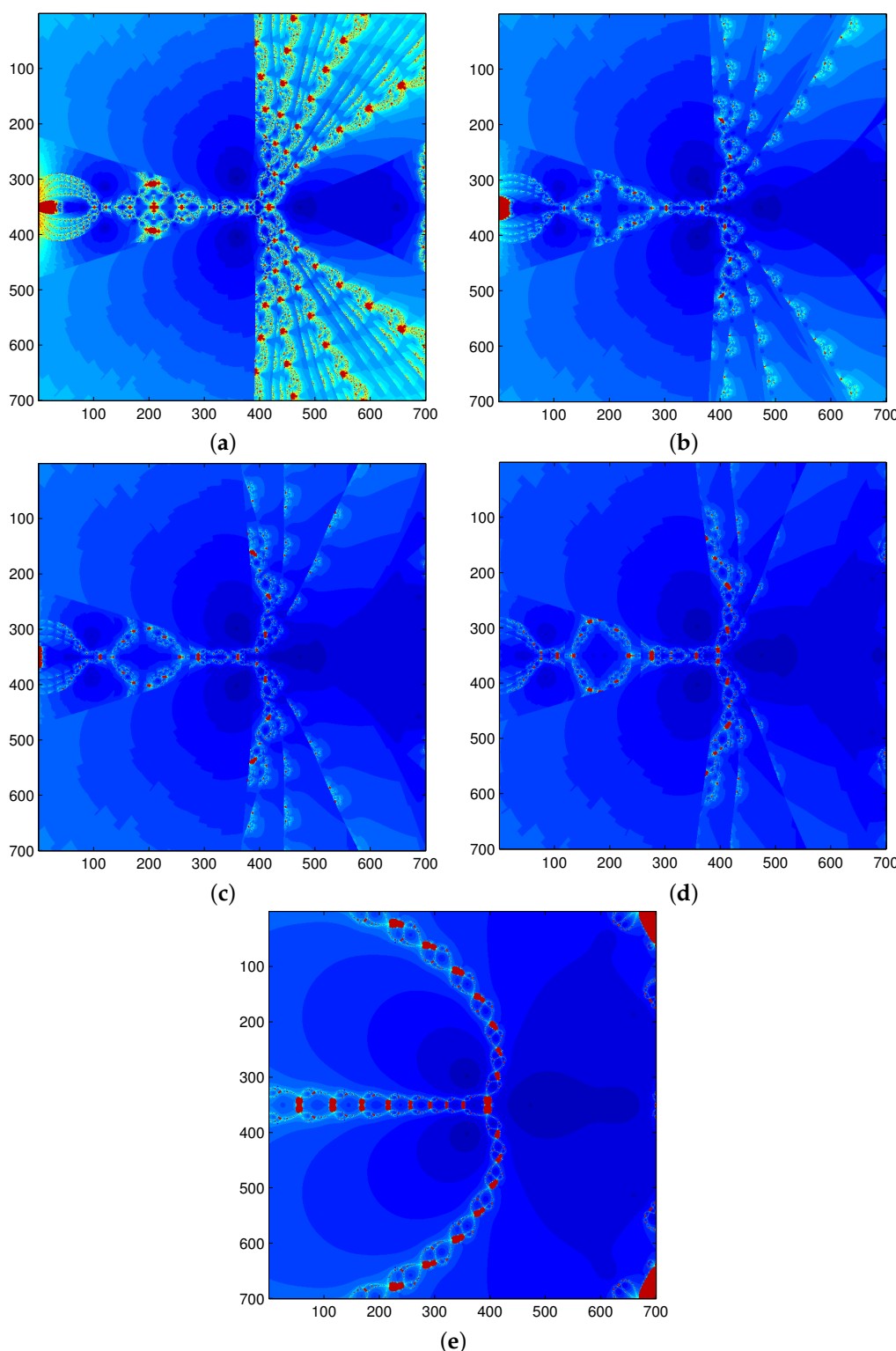

**Figure 8.** Dynamics of the proposed method with variable values of $\beta$ for $F_2(x) = x^3 - 2x^2 + x - 1$:
(**a**) proposed method at $\beta = 0.1$; (**b**) proposed method at $\beta = 0.3$; (**c**) proposed method at $\beta = 0.5$;
(**d**) proposed method at $\beta = 0.7$; (**e**) proposed method at $\beta = 0.9$.

## 7. Future Work

In future research, using our proposed method, we may consider the problem of
solving an algebraic equation that has roots with multiplicity. It will be interesting to see
the performance of a derivative-free version of the proposed method for the problems
considered in this study. The proposed method may also be discussed in Banach space to

solve the system of equations. Thus, the proposed method has indeed potential areas of interest that will be the topic of our future research.

## 8. Conclusions

We presented a new Newton-like normal S-iteration method for finding the root of the nonlinear equation $f(x) = 0$. Our theoretical results show that due to quadratic convergence, it requires only second-order differentiability rather than third-order differentiability. The numerical results and different graphic illustrations show that in spite of being a second-order convergence method, the proposed method is the most effective and superior when Newton's method fails, and it performs better than the same-order method of Fang et al. [11] and the third-order method of Wang and Liu [13], as it converges to the root much faster and very efficiently for different values of $\lambda_n$ with $\beta_n = 0.9$. Further, we showed that this convergence of the proposed method is accelerated for a sequence of variable values of $\beta_n$ converging to one. The results of the dynamic analysis also support the theoretical and numerical results related to the convergence and stability behavior of the proposed method. Thus, from a practical point of view, the new Newton-like normal S-iteration method has definite practical utility.

**Author Contributions:** Conceptualization, M.K.S. and I.K.A.; methodology, M.K.S. and I.K.A.; software, M.K.S., I.K.A. and A.K.S.; validation, M.K.S. and I.K.A.; formal analysis, M.K.S. and I.K.A.; investigation, M.K.S. and I.K.A.; resources, M.K.S. and I.K.A.; data curation, M.K.S. and I.K.A.; writing—original draft preparation, M.K.S. and I.K.A.; writing—review and editing, M.K.S. and I.K.A.; visualization, M.K.S. and I.K.A.; supervision, M.K.S. and I.K.A.; project administration, M.K.S. and I.K.A.; funding acquisition, M.K.S. and I.K.A. All authors have read and agreed to the published version of the manuscript.

**Funding:** This research received no external funding.

**Data Availability Statement:** Not applicable.

**Conflicts of Interest:** The authors declare that they have no conflicts of interest.

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
