# Peer review of "Newton-like Normal S-iteration under Weak Conditions"

_axioms, doi:10.3390/axioms12030283_

Round 1
Reviewer 1 Report
I would suggest a major revision based on the following comments and reasons.
1) The quadratic rate of convergence is not validated numerically. Please provide tables or plots to validate the theoretical rate of convergence. It may be helpful to show rate of convergence by plot.
2) I think the motivation for the present research would be clearer if the author could compare it with other available methods in terms of running time.
3) In the numerical example, all roots are simple. What if the root has a multiplicity?
4) Can you develop a derivative-free method?
5) Can you extend this method to system of equations? A comment would help the reader.
6) There are new references related to this work and should be cited. There are many related papers published after 2017 but the authors have not mentioned them. For example:
M. Baccouch, A Family of High Order Numerical Methods for Solving Nonlinear Algebraic Equations with Simple and Multiple Roots, International Journal of Applied and Computational Mathematics 3 (2017), 1119-1133.
Author Response
Author's Reply to the Review Report
Dear sir,
Following corrections have been made according to the reviewer’s comments
Reviewer 1
1. Sir, theoretical rate of convergence has been checked by numerical examples and numerical results have been supported by dynamical results.
2. Sir, running time is more or less equal.
3. Derivative-free method as well as method for system of equations will be very good topic for the future work. I have included a new section for it.
4. I have included the paper in reference.
Thanks with Regards

Reviewer 2 Report
Review manuscript ID: axioms-2009815
Type of manuscript: Article
Title: Newton’s like Normal S-iteration Under Weak Conditions
Authors: Manoj Kumar Singh, Ioannis K. Argyros*, Arvind K. Singh
Journal: Axioms [MDPI]
Date: 26 October 2022
The manuscript deals with a seemingly improved technique for solving nonlinear equations using a Newton-like method. Here are some remarks.
* It was mentioned that the third-order derivative is necessary, but then, in conclusion, it turned out that the third-order derivative is no longer important. Which one is correct? Please clarify!
* The authors claimed that their technique is better than the previous work, but it is still unclear what is new and what already existed in the body of literature.
* In connection with the previous remark, the introduction should be improved to cover the body of published literature that concerns the topic of discussion. Some important articles seem to be missing from the reference list, which includes but is not limited to,
Argyros, I. K., & Magreñán, Á. A. (2017). Iterative Methods and their dynamics with applications: A Contemporary Study. CRC Press.
Kotarski, W., Gdawiec, K., & Lisowska, A. (2012, July). Polynomiography via Ishikawa and Mann iterations. In International Symposium on Visual Computing (pp. 305-313). Springer, Berlin, Heidelberg.
Susanto, H., & Karjanto, N. (2009). Newton's method's basins of attraction revisited. Applied Mathematics and Computation, 215(3), 1084-1090.
Deng, J. J., & Chiang, H. D. (2013). Convergence region of Newton iterative power flow method: Numerical studies. Journal of Applied Mathematics, 2013.
Ardelean, G. (2011). A comparison between iterative methods by using the basins of attraction. Applied Mathematics and Computation, 218(1), 88-95.
The authors may want to check those references and include them accordingly to strengthen their literature study when discussing the complex dynamics of nonlinear equations.
* The symbol fraktur R for both real numbers and rectangles is dubious at worst and confusing at best.
* What is ANI?
* Line 160: manipulate your TeX code to avoid an overflow.
* Using i as an index might be confused with an imaginary number.
* Some figures blocked the caption.
* There are some problems with capitalization, boldface, italics, lonely bracket, hyphenation, etc.
Author Response
Author's Reply to the Review Report
Dear sir,
Following corrections have been made according to the reviewer’s comments
Reviewer 2
1. Conclusion related disorder have been corrected.
2. Body of literature has been improved.
3. Some important articles have been included.
4. Rectangles related typo has been improved.
5. ANI has been explained.
6. TeX code has been manipulated.
7. Remaining part of manuscript has been revised.
Thanks with Regards

Round 2
Reviewer 1 Report
The authors have taken much effort to improve the quality of the paper and also address all the reviewer's comments. I am overall happy with the modification, and would like to recommend the publication of this paper.
Some captions of the figures are hidden (specially Figures (b))
Author Response
Author's Reply to the Review Report
Round 2
Dear sir,
I am thankful to reviewer for the constructive and fruitful report of the manuscript.
Thanks with Regards
Reviewer 2 Report
Second review manuscript ID: axioms-2009815
Type of manuscript: Article
Title: Newton’s like Normal S-iteration Under Weak Conditions
Authors: Manoj Kumar Singh, Ioannis K. Argyros*, Arvind K. Singh
Journal: Axioms [MDPI]
Date: 18 November 2022
Thank you for revising the manuscript as well as responding to the previous inquiries. The manuscript looks much better now. Here are some additional remarks.
* From all the listed references, the authors mentioned the weakness of Newton's method. Some readers might wonder if there are some strong points among Newton's method, I am sure there are. Hence, why don't you improve your introduction to highlight not only the weaknesses but also the strength of Newton's method?
* Citation should be arranged according to its appearance in the text. The current version is jumping around and it is hard to follow.
* Previous remarks are not properly explained in the response letters, e.g.,
* It was mentioned that the third-order derivative is necessary, but then, in conclusion, it turned out that the third-order derivative is no longer important. Which one is correct? Please clarify!
* The authors claimed that their technique is better than the previous work, but it is still unclear what is new and what already existed in the body of literature.
* The italics problem still persist. As well as capitalization.
* Overall, many previous remarks have been ignored, including the fraktur notation.
* I copy pasted again the first review below.
~~~~~~~~~~~~~~~~~~~~~~~~~~~~~~~~~~~~~~~~~~~~~~~~~~~~~~~~~~~~~~~~~~~~~~~~~~~~~~~~~~~~~~~~~~~~~~~~~~~~~~~~~~~~~~~~~~~~~~~~~~~~~~~~~~~~~~~~~~~~~~~~~~~~~~~~~~~~
Review manuscript ID: axioms-2009815
Type of manuscript: Article
Title: Newton’s like Normal S-iteration Under Weak Conditions
Authors: Manoj Kumar Singh, Ioannis K. Argyros*, Arvind K. Singh
Journal: Axioms [MDPI]
Date: 26 October 2022
The manuscript deals with a seemingly improved technique for solving nonlinear equations using a Newton-like method. Here are some remarks.
* It was mentioned that the third-order derivative is necessary, but then, in conclusion, it turned out that the third-order derivative is no longer important. Which one is correct? Please clarify!
* The authors claimed that their technique is better than the previous work, but it is still unclear what is new and what already existed in the body of literature.
* In connection with the previous remark, the introduction should be improved to cover the body of published literature that concerns the topic of discussion. Some important articles seem to be missing from the reference list, which includes but is not limited to,
Argyros, I. K., & Magreñán, Á. A. (2017). Iterative Methods and their dynamics with applications: A Contemporary Study. CRC Press.
Kotarski, W., Gdawiec, K., & Lisowska, A. (2012, July). Polynomiography via Ishikawa and Mann iterations. In International Symposium on Visual Computing (pp. 305-313). Springer, Berlin, Heidelberg.
Susanto, H., & Karjanto, N. (2009). Newton's method's basins of attraction revisited. Applied Mathematics and Computation, 215(3), 1084-1090.
Deng, J. J., & Chiang, H. D. (2013). Convergence region of Newton iterative power flow method: Numerical studies. Journal of Applied Mathematics, 2013.
Ardelean, G. (2011). A comparison between iterative methods by using the basins of attraction. Applied Mathematics and Computation, 218(1), 88-95.
The authors may want to check those references and include them accordingly to strengthen their literature study when discussing the complex dynamics of nonlinear equations.
* The symbol fraktur R for both real numbers and rectangles is dubious at worst and confusing at best.
* What is ANI?
* Line 160: manipulate your TeX code to avoid an overflow.
* Using i as an index might be confused with an imaginary number.
* Some figures blocked the caption.
* There are some problems with capitalization, boldface, italics, lonely bracket, hyphenation, etc.
Author Response
Author's Reply to the Review Report
Round 2
Dear Sir,
Following corrections have been made according to the reviewer’s comments
Reviewer 2
1. The strength of Newton's method is added.
2. Citations have been arranged.
3. Sir, third-order derivative is important for third order method. It has been explained in the manuscript.
4. Body of literature has been improved.
5. Symbol fraktur R has been improved already in first revision. Symbols are different for fraktur and Real Number.
6. Sorry sir, I could not visualize the capitalization, boldface, italics properly. Please mention the specific. Whereas at some place they are for the purpose of clarity.
7. Remaining part of manuscript has been revised.
Thanks with Regards
Round 3
Reviewer 2 Report
Third review manuscript ID: axioms-2009815
Type of manuscript: Article
Title: Newton’s like Normal S-iteration Under Weak Conditions
Authors: Manoj Kumar Singh, Ioannis K. Argyros*, Arvind K. Singh
Journal: Axioms [MDPI]
Date: 29 November 2022
Thank you for revising the manuscript. Here are some minor remarks.
* The authors' response does not refer to particular line numbers or which portion of the text to indicate, which is hard to check.
* Use an improved, more compact citation style, e.g., [2-5], [6,7], [8-13], etc.
* The text starts with citation [2], where is the first one?
* Theorem as a general word should not be capitalized, the authors should learn again the basic rule of English capitalization.
* Why question was written in boldface?
* third-order. You have written this earlier. So, instead of me pointing one by one, I believe it should be the authors' responsibility to search for some mistakes and consistencies in the text.
* Newton's method.
* sin and cos should be written in math mode.
* Figure labeling is too small. Please enlarge it.
* Some figures were blocking the other figures. Please format them carefully.
* boundaries (again, these mistakes should be the authors' responsibility to find, not mine.)
* Reference style is still not consistent.
Author Response
Author’s Report
Dear sir,
Please find the revised manuscript for possible publication.
compact citation style is adopted
Citations are corrected
Theorem as a capitalization is improved
Question was written in boldface is corrected
Third-order and other this type of typos are improved
Newton's method and other this type of typos are improved
sin and cos are improved in math mode now.
Figure’s labeling is corrected.
Reference style is corrected.
The manuscript is modified and corrected according to the reviewer’s comments

Round 4
Reviewer 2 Report
Fourth review manuscript ID: axioms-2009815
Type of manuscript: Article
Title: Newton’s like Normal S-iteration Under Weak Conditions
Authors: Manoj Kumar Singh, Ioannis K. Argyros*, Arvind K. Singh
Journal: Axioms [MDPI]
Date: 24 February 2023
The paper presents a new quadratically convergent Newton-like normal S-iteration method for solving nonlinear equations that may have points in the neighborhood of the root where the first derivative equals zero. The authors claim that their method outperforms the Fang et al. method, and provide numerical and dynamical results to support their claim. However, the paper lacks clarity and precision in explaining the technical details of the method and the mathematical proofs of its convergence, divergence, and stability. The writing style could be improved to make the paper more accessible to a wider audience.
Thank you for revising the manuscript. I appreciate your efforts. I do have a few minor remarks to make, however.
* Some panels in Figures 8 blocked the captions.
* Some panels in Figures 7 also blocked the captions.
* Some panels in Figures 6 blocked the captions.
* Some panels in Figures 5 also blocked the captions.
* Some panels in Figures 4 blocked the captions.
* Some panels in Figures 3 also blocked the captions.
* Some panels in Figures 2 blocked the captions.
* Some panels in Figures 1 also blocked the captions.
* A suggestion for an improved abstract:
This paper introduces a new method for solving nonlinear equations, called the quadratically convergent Newton-like normal S-1 iteration method, which does not require the second derivative and can handle cases where f ′(x) = 0 in the vicinity of the root. Our proposed method outperforms the Newton method and some higher order converging methods, especially in cases where Newton's method fails. Numerical results demonstrate that our method converges faster than the method introduced by Fang et al. (J. Compute. and Appl. Math., 220 (2008), 409-412), which is cubically convergent under weak conditions. We conducted a thorough investigation of the normal S-iteration method to identify aspects that contribute to its faster convergence to the root. Finally, our dynamical results confirm the numerical results and offer an explanation for the convergence, divergence, and stability of the proposed method.
Author Response
Dear sir,
Manuscript has been revised according to your comments. Please find it for possible publication.
Thanks with Regards